# Patients’ with Multimorbidity and Psychosocial Difficulties and Their Views on Important Professional Competence for Rehabilitation Coordinators in the Return-to-Work Process

**DOI:** 10.3390/ijerph181910280

**Published:** 2021-09-29

**Authors:** Azadé Azad, Veronica Svärd

**Affiliations:** 1Department of Clinical Neuroscience, Division of Insurance Medicine, Karolinska Institutet, 171 77 Stockholm, Sweden; 2Department of Psychology, Stockholm University, 106 91 Stockholm, Sweden; 3Department of Social Work, Södertörn University, 141 89 Huddinge, Sweden

**Keywords:** return to work, rehabilitation, coordinator, professional competence, qualities, skills, knowledge, patient perspective

## Abstract

Coordinators may play a key role during the return-to-work (RTW) process for people on sickness absence. There are still few studies on the newly implemented rehabilitation coordinators (RECO) within Swedish healthcare, and none focus on their competence. The aim of this study was to explore how persons with multimorbidity and psychosocial difficulties describe the professional competence of the RECO they encountered during their RTW process. The study takes a relational and practical approach in defining professional competence, including both what professionals do and what they possess. Interviews with 12 people with multimorbidity and psychosocial difficulties who had encountered a RECO during their RTW process were analysed using thematic analysis. Six different themes were found: communicative and coordinating skills; advisory and guidance skills; engagement and advocacy skills; being persistent and flexible; being empathic and therapeutic; being professional and trustworthy. Most of these are found in research on RTW coordinators, but being persistent, and having advisory, guidance, advocacy and therapeutic skills have not been recognised as important competences previously. This study adds patients’ views on important professional competence that support the RTW process, which should be regarded in further developments of RECOs’ functions and their competence descriptions.

## 1. Introduction

The role of coordinators as professionals who play a key role during the return-to-work (RTW) process after sickness absence (SA) has been the focus of increasing research in recent years. In the literature about coordinators in the RTW process, these professionals are often called RTW coordinators, although the terms used differ. Internationally, RTW coordinators often work within the workplace or at insurance companies, but in Sweden they are mostly placed within the healthcare system, where they are called rehabilitation coordinators (RECO). Although RTW coordinating services are still in their early stages in Sweden, they are rapidly increasing, due to the ongoing and widespread implementation of RECOs. Since 2020, healthcare services in Sweden are obliged to offer patients on SA rehabilitation coordination if needed [1]. The function of the RECO is to give individualised support to the patient in the rehabilitation and RTW process, as well as to initiate and manage both internal coordination with other healthcare professionals and external coordination with, for example, the employer, the Social Insurance Agency (SIA), and the social services [2]. They are not restricted to working with people with work-related injuries, as may be the case in other countries; rather, RECOs work with all types of medical diagnoses resulting in work disability and SA, regardless of what causes any individual diagnosis. Previous reviews on the effects of having support with coordinating the RTW process have shown mixed results: both moderate [3,4,5] and a lack of effects [6,7] on different parts of the RTW process have been reported. The effects of RECOs in primary healthcare in Sweden have also been evaluated, with studies showing positive outcomes, for example, in decreased SA days and a faster RTW process [2,8]. However, the reasons why and how RTW coordination is successful or not are still unclear, and studies exploring the importance of professional competence in this function are few.

Early research by Shaw and colleagues [9] described the activities of RTW coordinators and case managers in published trials with the purpose of providing a basis for establishing necessary competencies, and they identified six different domains of competencies for RTW coordinators, namely workplace assessment, clinical interviewing, social problem solving, workplace mediation, knowledge of business and legal aspects, and knowledge of medical conditions. In a follow-up study by the same research group [10], the principal investigators for intervention studies on RTW coordinators in several countries were interviewed, and they found ten groups of essential competencies they deemed as important: individual traits/qualities, relevant knowledge base, RTW focus and attitude, organisational/administrative skills, assessment skills, communication skills, interpersonal relationship skills, conflict resolution skills, problem-solving skills, and RTW facilitation skills. Based on RTW coordinators’ participating in focus group interviews and a survey, the same research group [11] added professional credibility and information gathering as essential competencies. 

Other researchers have further elaborated on this work and identified personal skills such as gaining people’s trust, being positive and having good conflict-resolution skills [12], as well as good communication skills and empathy [13]. Workplace RTW coordinators themselves have identified knowledge in medical terminology, counselling skills and appropriate record keeping skills as necessary to assist in facilitating the RTW process [14]. A lack of certain traits and skills has also been demonstrated; for example, Bohatko-Naismith and colleagues [15] found that RTW coordinators who were perceived as inexperienced and unsupportive or who demonstrated poor communication skills could act as a barrier in the RTW process. As previous studies have investigated the competencies of RTW coordinators from the viewpoint of RTW coordinators themselves [11,12,14,16], other stakeholders [13], or experts [10], there is a lack of knowledge as to what patients view as important skills for the coordinator. One study [17] exploring injured workers’ experiences with their workplace RTW coordinators and the barriers the workers may have encountered during the RTW process, found that listening and communication skills, empathy, engagement, and knowledge of legislation were important, and found barriers such as RTW coordinators’ being unsupportive, inexperienced and apathetic.

It is likely that the RTW coordinating competences outlined are important for RECOs to possess as well; however, this has not been studied to date. As others [13] have pointed out, international comparisons of various coordinators involved in the RTW process in general may be difficult due to variations in both the titles, tasks, national legislation and social insurance systems, as well as in what sector they are employed. The different contexts of where coordinators are situated would seemingly require different competence.

### 1.1. The Competence Description for Rehabilitation Coordinators

Even though Swedish healthcare services are obliged to offer rehabilitation coordinating services, there are only vague descriptions of the required competence for this function. The lawmakers found no reason to enforce any regulation about this occupation or its minimal qualifications, leaving this decision to healthcare managers. Most often, a RECO will have a professional background and work experience in healthcare as, for example, a registered physiotherapist, occupational therapist, nurse, or health social worker. The Swedish Association of Local Authorities and Regions (SALAR) have published a method book for RECOs [2], in which it is stated that a RECO should have knowledge about insurance medicine, social insurances, the rehabilitation process, medical terminology, assessments made in healthcare and insurance medicine, the labour market, work-related laws and rehabilitation efforts, and gender equality. Depending on the setting—for example, primary or specialist healthcare—the RECO is expected to have knowledge about different diseases and health conditions. General qualities are also mentioned, for example, good ability for interpersonal relationships, good communication skills, and good organisational and administrative abilities, which also include problem-solving skills. Whether or not these skills and qualities are those which are regarded as important from a patient perspective are, however, unclear. 

However, related studies on how people on SA experience the functions and support provided by RECOs point at several functions as important, such as enhanced communication and collaboration between different stakeholders involved in the RTW process [18,19,20]. Additionally, RECOs tend to base their work practice on the basis of the competence acquired through their vocational education and professional background, where the diversity of professional backgrounds and training among RECOs leads to variations in how coordination is conducted [20]. RECOs have also been described as providing structure and giving practical and mental support and encouragement [18,21]. For persons with multimorbidity, RECOs have been described as providing value and offering support in several areas of the rehabilitation process by, for example, coordinating contacts and offering advice with healthcare professionals and employers [18]. Although these studies may say something about the areas of importance for RECOs to be knowledgeable in, no study has to the best of our knowledge specifically focused on the competence of RECOs. In order to develop the RECO’s role and a description of the required professional competence, such studies are of importance.

### 1.2. Defining Professional Competence

The concept of professional competence is understood in numerous ways. The above-mentioned studies on competence among RTW coordinators mostly used a positivistic approach, considering competence as entailing individual resources [10,11] or as a cognitive ability of knowing how to act [16], or otherwise did not define the concept of competence at all [9]. The present study is based on a relational theoretical approach that considers the practical and contextual dimension of competence, and views competence as what is used by professionals in the interactions with others in a specific context, looking at what they do as well as what they possess [22]. We are inspired by Lindberg and Rantatalo’s [23] operational definition of professional competence as:
the inferred potential for desirable activity within a professional practice. This definition also entails that desirability of any activity is a continuum, where excellence is relationally constructed against what is merely approved or suitable. [23] (p. 565)

To further operationalise this definition, they suggest that professional competence can be viewed as comprised of qualities (traits, skills) and symbols (appearances, feelings, activity in practice), where the symbolic manifestations, such as what is considered as good social competence, can differ considerably with certain contexts. This relational approach is appropriate when analysing what descriptions of competence emerge in patients’ descriptions of RECOs, because what is perceived as competence is constructed within their very relations. Further, the relational approach considers knowledge as a tool that enables practice but possessing knowledge does not naturally lead to acting competent and knowledgeable. The focus of the present study is not only to study what knowledge the participants believed the RECOs had or did not have, but also to look at how this knowledge was conveyed to the participants in practice.

### 1.3. Aim of the Study

The present study is based on interviews with people with multimorbidity and psychosocial difficulties who encountered a RECO during their SA and RTW process. This is a group that has been suggested to significantly benefit from coordination during the RTW process, as the rehabilitation of people with multimorbidity often involves challenges and obstacles [18,24]. The aim here is to explore how persons with multimorbidity and psychosocial difficulties describe the professional competence of RECOs they encountered during their RTW process.

## 2. Methods

This study has a qualitative approach and used semi-structured individual telephone interviews. Due to the COVID-19 pandemic prohibiting in-person interviews during the data collection period, we used telephone interviews. Thematic analyses were used, a method of analysis that is frequently used within healthcare research, and well suited to inductive analysis and flexible, as it is not bound by a pre-existing theoretical framework [25]. 

### 2.1. Participants

The inclusion criteria were people who had two or more diagnoses and/or social difficulties hindering their capacity for rehabilitation and returning to work, and who, during their SA, had contact with a RECO. The participants were 12 people, nine females and three males, and they were between 34 and 50 years of age, living in Region Stockholm, Sweden. Sociodemographic information about the participants is presented in Table 1. They all had at least two diagnoses, and a majority also described psychosocial difficulties affecting their symptoms, rehabilitation, or RTW process. The most frequent occurrent diagnoses were common mental disorders such as mental fatigue symptoms and depression, but somatic diagnoses such as cancer also occurred. Further details about the diagnoses and psychosocial difficulties are found in a separate study [18]. The majority of the participants were working part-time at the time of the interview, and were or had been on SA between 6 months up to approximately four years. Most were still in contact with a RECO at the time of the interview, and they all had had several contacts with a RECO during their SA.

### 2.2. Recruitment

Thirteen RECOs working at primary healthcare, psychiatry or addiction centres, were asked by the project leader (VS) to send a letter about the study to three to six eligible patients each (in total 70 patients), preferably with different genders, backgrounds, and experiences. Due to the Swedish Secrecy Act, it was not possible for the project leader to obtain any information about those invited. The information letter, which was sent out once, contained a short background summary of the study, voluntary participation, contact information to the researcher for further questions, a consent form, and a pre-paid response envelope. Thirteen persons replied with a signed consent form together with their contact information, and they were then contacted by the interviewer (AA), informing them verbally about the study and scheduling a time for an interview. One individual who sent in a consent form declined to participate when approached by the interviewer, due to having insufficient knowledge about their RECO. 

Before the interviews started, the participants received written and oral information explaining that research participation was voluntary, and that their participation and data would be treated with strict confidentiality. They were also informed that they could withdraw at any time without consequences, and were assured of their anonymity in the handling of the data. The researchers received no information about those who were invited, and the RECOs have no information about the final participants.

### 2.3. Interviews

The telephone interviews were conducted from May through August 2020 and lasted between 30 and 74 min each. A semi-structured interview guide was developed based on previous literature on RTW coordination. The interview guide addressed how the participants experienced and viewed their SA, and how living with multimorbidity and psychosocial difficulties affected their rehabilitation and RTW process. The participants were also asked to describe the contact with their RECO, how they had experienced the support received from the RECO, and how valuable they deemed the RECO’s support had been in their RTW process. A previous study focusing on the more direct answers to the interview questions, by exploring barriers in returning to work due to the participants’ multimorbidity and psychosocial difficulties, and the support received from RECOs in different areas of the RTW process, is published elsewhere [18]. Regarding the present study, however, there were no specific interview questions addressing professional competence; instead, we explored how the participants during the interviews described the professional competence of the RECO they encountered. The interviews were conducted in Swedish and recorded and transcribed verbatim. We asked the participants for permission to record the interview, and they all agreed. All names, personal information and places were omitted and/or changed in the transcriptions as well as in the write up of the study. 

### 2.4. Coding and Analysis Process

Thematic analysis [25] was used for analysing the interviews, and coding was done inductively. Two researchers, AA and VS, conducted the analysis by initially reading all 12 interviews to get an overall impression of possible themes in the data. Next, meaning-bearing units (sentences and phrases that contain relevant content with regard to the aim) were identified based on the manifest content and condensed and rewritten. After that, AA looked for common patterns in order to identify preliminary themes. This was done by abstracting and coding the condensed units corresponding with the aim. VS then read the first draft of the themes which were jointly discussed in relation to the theoretically informed concept of professional competence, and revised these until a consensus was reached [26]. The final step was naming the themes and assigning units to each theme. For validation, the themes were double-checked with the raw material (i.e., the interviews with each participant). This was done until the conceptual depth in the themes was agreed upon [27]. 

Both authors read the transcribed interviews several times, and continuously discussed the coding and themes during the analysis process. These discussions continued until a consensus was reached and inter-coder reliability was established [26]. We believe that the way both authors became very familiar with the interviews and repeatedly discussed the codes and themes, strengthened the credibility of the analysis process.

## 3. Results

The analysis of what was deemed as important professional competence of RECOs resulted in six different themes. These are elaborated on below with quotations for each theme. One specific competence that was consistent in all themes was communicative skills, which in different ways was emphasised throughout the different themes. The themes relate to both symbols, such as appearance and activity in practice, as well as general personal qualities or specific skills related to specific situations or issues. A theoretical discussion is further presented in the discussion. 

### 3.1. Communicative and Coordinating Skills

One theme included communicative and coordinating skills, which were considered important for all the participants during their RTW process. These included RECOs helping them keep track of the various instances they were in contact with when necessary and facilitating communication. 

When I haven’t been able to reach my physician/…/I’ve been able to call [the RECO] and then check what she feels and thinks. And then she’s been able to talk directly to my physician, and they’ve reached an understanding. And then she calls me back. (Participant 1)

Communication with physicians, the SIA, and employers was regarded as vital when the participants felt that, due to their illness, they were not able to maintain these contacts in a productive way on their own.

Other qualities having to do with coordination included administrative skills, such as being punctual and direct and being good at organising and planning. One participant linked coordinating skills to a sense of security:

I think that for me it’s been mostly a bit of security that she’s [the RECO] someone who keeps track of the various instances. Yes, more like a sense of security than anything else.(Participant 2)

### 3.2. Advisory and Guidance Skills 

Another competence that the participants described as important was that RECOs were able to give advice, for example, about how to express their symptoms, needs, and work disabilities to employers or physicians or other stakeholders, as well as advice about rights, or possible workplace measures. It also included advice and guidance within the healthcare system, such as choosing a suitable physician, as expressed by one of the participants: 

Then she could support me like “okay but now I think we should take this doctor because he, yeah well, he’d be better suited for you, and he’s responsive.” (Participant 11)

It was also appreciated when RECOs offered advice and guidance about economic solutions or things that fell outside what the patient was on SA for, for example, other social problems in life. One participant expressed this by saying: 

If someone comes to you and shows how to think, this is how you do it, you should eat this, you should do this, that’s worth so much. (Participant 8)

However, not all participants received this type of support from their RECOs, although they wished they had. 

The participants also considered that it was important that the RECOs offered advice on how to adjust the RTW in terms of a slower or faster RTW pace. Receiving tailored advice from RECOs on their RTW pace was considered important for a successful RTW process. For those who did not receive help with and advice on a slower pace when they would have needed this, they had become more ill and had a harder time returning to work in a sustainable way.

### 3.3. Engagement and Advocacy Skills

Advocacy skills were found as important, exemplified by the RECOs representing the participants’ points of view at meetings with other stakeholders, making sure their voices and wishes were heard, as well as clarifying the participants’ right to rehabilitation measures. One participant said: 

I’ve felt that the rehabilitation coordinator has somehow really been on my side during this process, so to say. But I guess it’s because counsellors and such don’t tend to put diagnosis on you, rather just trying to help. (Participant 4)

RECOs who were described as having advocacy skills also led to participants receiving better encounters from other professionals and stakeholders and feeling more empowered and less alone in the RTW process. 

This theme also included descriptions of RECOs being compliant with the participants’ different needs during the different phases of the RTW process, and being engaged and interested in their situation, a competence more related to how RECOs appeared during their contact. One participant described this engagement by saying: 

She actually called me at home the evening before I was gonna go to this meeting with this doctor. She gave me a pep-talk on what to say. (Participant 6)

A positive and supportive attitude towards the participants’ abilities was thus important, as well as strengthening their voice and rights in relation to other stakeholders.

### 3.4. Being Persistent and Flexible

This theme, being persistent and flexible, is related to symbols, as it refers to RECOs’ appearance and their activity in practice. Many regarded RECOs’ persistency and flexibility as important, which had to do with having continuous contact with someone involved in the RTW process from the beginning to the end. This made the participants feel like they were taken care of within a system in which they mostly lacked trust. In this regard, RECOs were the ones within the healthcare system who the participants had the closest and most frequent contact with. Being persistent also included the RECOs calling the participants on a regular basis to check in and offer feedback on their SA process. Being persistent involved the RECOs not giving up in difficult situations, but rather offering alternative strategies or giving the extra push needed, as described by one of the participants: “She genuinely seems to care and she doesn’t let go” (Participant 12). 

Being flexible was also deemed as valuable. For example, one participant said: “when you meet her, or just on the phone, she always has time to talk” (Participant 6). In this sense, it was important that the RECO not only had time to talk, but that the time of contact was not limited, in contrast to, for example, physicians, who offer only very limited time periods at doctors’ appointments. In the same way, not being able to reach the RECO, or the RECO not returning their phone calls, was described as difficult, with one participant saying: “She still hasn’t called. It’s been problematic” (Participant 7). Being flexible also meant that it was important that the RECOs were flexible in their schedules so that they could attend meetings with other stakeholders on short notice. In the same way, participants described negative consequences when their RECO was not able to attend collaborative meetings with the patient.

### 3.5. Being Empathic and Therapeutic 

A RECO was described by the participants as someone not just to talk to about their SA situation, but also about difficulties in their lives in general. In that way, having contact with a RECO was, for the participants, in some way therapeutic. For some participants, their RECO thus had the role of a mental coach, counsellor or psychotherapist. In line with this, competence having to do with communication skills and empathy was important, that is to say, a competence in more symbolic terms, meaning that the RECO conveyed and evoked positive feelings. For example, participants stressed that having a RECO that was empathic and easy to talk to, or who listened and was calming and could help them sort out their thoughts and emotions, was important. 

Or if I’m going somewhere, I’ve asked… Can we talk before? Then. Yes, so I can sort out… what’s important. Because there’s so much emotions. She has helped me sort out like and… Yes, bring clarity to all my thoughts and how I should speak, and what I should say, and… Yes. So that she… Yes. She has been a very good support. (Participant 5)

Other participants described that the RECO could provide them with new insights and points of view, and possibly change the way they thought. The participants also described that they felt secure when talking to their RECO, as this was a person with whom they could be totally honest, in contrast to, for example, physicians, employers or SIA officers. Having someone to talk to who was non-judgmental was of great importance. The way empathy could be important was described by one participant, who had had a hard time describing her difficulties to healthcare professionals; she appreciated the fact that her RECO could see through her external appearance and actions:

I feel that it’s on a very personal level actually. I feel that some people in healthcare immediately get my… I try. I’m aware that I tend to kind of ease over things and be a little too… well, I don’t know how to describe it. Bubbly and like, “oh, it’s fine”. But some [the RECO] seem to see through it. (Participant 4)

### 3.6. Being Professional and Trustworthy

Other important competence had to do with symbols, such as the RECOs’ appearing professional, knowledgeable and trustworthy. A RECO’s ability to clearly present their professional role and mission was deemed as important; participants who did not receive such information consequently were unsure of how to evaluate their RECOs’ competence to uphold their mission. From the patient’s perspective, it was important that the RECO took her/his profession seriously, was meticulous, and had knowledge about legal rights and responsibilities, their specific diagnoses, rehabilitation measures, prognostic factors and realistic goals for recovery, while at the same time being able to have an overview of their whole life situation, including their multimorbidity and other social circumstances and barriers. One participant added that knowledgeability involved understanding the specific process: 

She seems to take her profession seriously/…/She’s knowledgeable about this/…/like understands the process in this type of sickness absence. (Participant 12)

Being professional and trustworthy was also about their activity in practice, as this included RECOs making an effort to learn about things they themselves lacked knowledge about. For example, when a participant had an unusual disease which their RECO at first did not have knowledge about, she read up on the disease, what it means, and how to think about it. Being knowledgeable as an aspect of their professionalism was regarded as important, with one participant saying: 

For me the most important thing has been that she’s so knowledgeable in how fatigue syndrome works and has experience with it. When I thought I was going crazy, she was able to tell me that “no, that’s perfectly normal. That’s how it is”. And that’s been my salvation. (Participant 5)

In the same way, other participants described that when their RECO did not have certain knowledge, about either medical conditions or the legal system, the participants felt this hindered their RTW process.

## 4. Discussion

The present study analysed how patients with multimorbidity and psychosocial difficulties described the professional competence of the RECO they encountered during their SA and RTW process. Our purpose was not to define the right professional competence, or the right competences for a RECO to have; instead, our focus was to explore the patient perspective on this. The findings add to previous literature, for example, [9,10,11,12,13,14,15,16,17] by pointing to six different themes of competence, including both general and specific qualities as well as more symbolic aspects of competence, such as how the RECO appeared or acted. The six themes found were: communicative and coordinating skills, engagement and advocacy skills, advisory and guidance skills, being empathic and therapeutic, being persistent and flexible, and being professional and trustworthy. While some findings, such as communication and coordination, are more clearly defined as skills, other findings, such as being empathic, professional and trustworthy, have more to do with what Lindberg and Rantatalo [23] defined as symbols, as they relate to how the RECOs appeared and acted or what feelings they evoked. Professional competence, as performed in the very relation between a RECO and a patient, is thus vital from a patient perspective.

The professional competence described included the domains of knowledge, skills, personal characteristics, and symbols, suggesting that RECOs having a combination of these aspects is important during the RTW process. The combination of interpersonal as well as knowledge-based competence suggests a duality in patients’ views of competence, in line with the theory of the duality of competence and caring [28]. This theoretical model was initially developed with respect to patients’ views of nurses’ competence, suggesting that capability in knowledge and skill are deemed as equally important as interpersonal skills such as relating to the patient in an empathic way. 

Our findings complement and, in some ways, contradict earlier studies on RTW coordinator competencies, both regarding empirical findings and theoretical discussions. For example, whereas Pransky et al. [11] and Gardner et al. [10] in their definition of competence used individual resources, and Durand et al. [16] applied a cognitive approach, we defined competence as not only possessing theoretical knowledge but also to actively put this knowledge into practice when encountering a patient. Putting knowledge into practice, for example, involves giving advice about regulations so that patients understand them, or guiding patients in how to express their symptoms, needs, and work disabilities to employers or physicians or other stakeholders. We therefore suggest that when studying professional competence of RECOs, more emphasis should be placed on not only what is, but also what is shown, expressed and practiced. While specific knowledge about, for example, diseases or the social insurances was perceived by the participants as important to promote the RTW process, the relational and practical aspects of competence call for a broader definition of competence. While “possessing” knowledge about, for example diseases or the social insurances is an important prerequisite for successful practice, the doing-of-knowledge, meaning how to put knowledge into practice, was regarded as equally important, as this lays down the conditions for making use of knowledge. In line with this, the relational aspects of the practice, outlined by Lindberg and Rantatalo’s [23] definition of professional competence, are vital. Some competence identified in the present study, such as factual knowledge about, for example, legislation, may be obtained through formal or informal education and training. However, other competence such as engagement, empathy and advocacy can be characterised as symbolic competence that benefit from practical training or certain personalities, and may not as easily be learned.

Regarding the empirical findings, Pransky et al. [11], in line with our findings, reported professional credibility and administrational skills as important, and being knowledgeable about legal aspects as well as medical conditions have been reported as important by Shaw et al. [9]. Bohatko-Naismith et al. [13,17] found, similar to our findings, that good communication skills and empathy were important characteristics/traits to have. Some of the highest ranked competencies from Pransky et al.’s [11] study (e.g., being committed to the goal of an early RTW) were not mentioned by our participants, while some of their lowest ranked competencies (e.g., taking a client-centred approach and the ability to evaluate potential impacts of co-morbidity) were important findings in our study. There are several possible explanations for this, with one being that the participants in our study had multimorbidity. Another explanation could be that the role (and thus the professional competence) of a RTW coordinator in another country is not always comparable to the role of a RECO in Sweden due to contextual or situational factors affecting expectations. One explanation is also, importantly, different perspectives (professionals, experts, patients/clients) on what constitutes important competence. Most empirical studies on professional competence, both regarding RTW coordinators and other professionals, are based on professional or managerial views, and the present study thus adds the patient/client view on what constitutes good professional competence for RECOs. However, as they each contribute to the compilation and qualification of important competence from different perspectives, the different approaches are each legitimate. 

A recent systematic review [3] of the impact of RTW coordinators also found that there are few studies focusing on how patients experience RTW coordination, and how these aspects affect the RTW process. Although our study does not speak to outcomes, the findings do suggest, from a patient perspective, the importance of professional competence when encountering a RECO during a RTW process. This could guide future studies on why some coordination interventions are beneficial while others are not. Further, from a patient perspective, it may be important to explore both what is practiced and what is not. Our findings reflect the importance of not only competence that is shown or practiced, but also what is lacking. A lack of competence and certain skills can be perceived as equally decisive as the presence of them in the RTW process [13,17]. In our findings, a lack of advisory and guidance skills, both regarding health conditions and social circumstances, as well as a lack of persistency and flexibility, i.e., not being able to contact or have the RECO present during meetings with stakeholders, were regarded as hindering the RTW process.

The present findings have implications for developing the role of the RECO. In SALARs’ method book for RECOs [2], both general and specific competence and skills are mentioned, of which many are in line with our findings. These include knowledge about different diseases and health conditions, social insurances, and the rehabilitation process, as well as good communication and organisational skills and administrative ability. However, advisory and guidance skills are not mentioned; neither are being persistent or having engagement, advocacy or therapeutic skills, although a good ability for interpersonal relationships is mentioned [2]. Taking a patient perspective, these forms of competence should be considered to inform descriptions of RECO competence.

The study involves methodological limitations. As the sample size was small, and all the participants resided in one urban setting, the generalisability of our findings may be limited. However, the RECOs were employed in different settings within both primary and specialist healthcare services, which offers more diversity. As the RECOs were the ones who selected the patients eligible to be invited to participate in the study, and as we as researchers had no information about those who were or were not invited, we cannot judge possible biases. There could therefore have been differences between those who were invited or not, as well as differences between those invited who chose to participate or not. Patients with negative attitudes or experiences of RECOs may have declined participation, along with patients with little knowledge about RECOs or their mission, as was the case for one person, who declined to participate when approached. However, the participants who were interviewed described various types of experiences, both positive and negative, suggesting a diverse range of experiences of those included. 

This study adds knowledge about patients’ perspective of important competence for the coordinator role, a perspective rarely included. Furthermore, it included a group of patients with multimorbidity and complicating psychosocial difficulties, a group of which there is limited knowledge about regarding their experiences during the RTW process. This specific group of patients adds to previous research, but the results should also be viewed through the specific needs they have, due to their multimorbidity and psychosocial difficulties, which may not always be generalisable to other groups of patients. Further studies could benefit from more closely investigating if and how different fields of practice (i.e., contextual factors) may affect the competence perceived as important, both from the perspectives of patients and RECOs themselves. It would also be valuable if future studies investigate the needs of patient groups with different diseases and difficulties.

## 5. Conclusions

In conclusion, the findings of this study provide a unique insight into professional competence, including a range of qualities, skills and knowledge that patients with multimorbidity and psychosocial difficulties describe as important for RECOs to have in order to facilitate the RTW process. The professional competence described can be used to inform description of the required competence for RECOs within Swedish healthcare, as well as informing selection criteria for individuals choosing to enter this field.

## Figures and Tables

**Table 1 ijerph-18-10280-t001:** Sociodemographic characteristics of the participants (*n* = 12).

Characteristics	*n* = 12
Gender	
Female	9
Male	3
Age	
Median years (range)	48 (34–50)
Sickness absence	
Full-time	3
Part-time	8
Returned to work	1
Duration of sickness absence	
Median years (range)	1.5 (0.5–4)
Occupation	
Office work	9
Manual labour	2
Unemployed	1
Contact with RECO through	
Primary care centre	7
Psychiatric or addiction clinic	5
Number of contacts with RECO	
More than 6	8
Up to 6	4

## Data Availability

The data presented in this study are available on request from the author V.V. The data are not publicly available due to ethical restrictions.

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
