# Peer review of "Patients’ with Multimorbidity and Psychosocial Difficulties and Their Views on Important Professional Competence for Rehabilitation Coordinators in the Return-to-Work Process"

_ijerph, 2021, doi:10.3390/ijerph181910280_

Round 1
Reviewer 1 Report
Thank you for the opportunity to read the manuscript. I think the study is interesting and a valuable contribution developing the competence of RECOs in the Swedish healthcare system. The manuscript is well written but I have found some things that could improve the manuscript. One important comment is to declare if the study has an ethical approval. I will also suggest a language editing of the manuscript. You find my comments below.
Title: I would like some more information of the study in the title, for example that the participants/patients in the study are persons with multimorbidity and that the context for the study is Sweden. One suggestion I have is: Patients’ with multimorbidity and their view of professional competence for rehabilitation coordinators in the Swedish healthcare system. Or find an appropriate subtitle that could complement with information about the patient group and context the study is made in.
The aim is described differently in different parts of the manuscript, in the abstract, the introduction and in the discussion. Please look through and decide the aim and then write the aim with the same wording/meaning when it is mentioned.
Another important comment is that I would like the authors to decide, and be consistent when they refer to and write about rehabilitation coordinators. The rehabilitation coordinator has different names depending on which countries and what contexts they operate in. Therefore, it is important to be clear when different notions are used. In the paper, the authors mix different notions throughout the paper, which makes it difficult for the reader to follow. For example, the authors use coordinators, rehabilitation coordinators, RTW coordinators, RECOs, and so on. When the different notions are used, it has to be clear for the reader when and why different notions are used, and also to be consistent in when to use the different notions. For example, when the rehabilitation coordinator is mentioned by the participants, or if the author refers to a Swedish context, RECO could be used.
Abstract: The abstract may have to be changed after revisions, in this version, the aim is not clearly described.
- Introduction:
In the beginning of the introduction, line 33, after the first sentence, one suggestion is that the authors could explain that most of the studies about coordinators in the RTW process mostly are called RTW coordinators, and include in what context these coordinators are found (for example in workplaces, Europe, worldwide, etc). Thereafter, it could be appropriate to explain the role of the rehabilitation coordinator in Sweden, and introduce the shortening RECO. Another suggestion is to move the sentences about the Swedish context and RECOs to the end of the introduction, to be included in the last paragraph (line 77) before the theoretical framework.
Section 1.1. The theoretical framework is very interesting. But perhaps it could be better to present the theory after the description of RECOs in the Swedish context. In other words, change 1.1 to become 1.2 and vice versa.
Section 1.3 Aim of the study – as mentioned earlier, the authors have to decide and become clearer and more consistent in describing the aim of the study. For example, I perceive that the aim of the study is something like, investigating/exploring how persons with multimorbidity and psychosocial difficulties describe/experience the professional competence of a rehabilitation coordinator (RECO) when needing support in a return-to-work process.
- Method
2.2 Recruitment.
In this section I would like to have a description of inclusion and exclusion criteria. I also lack information if the study has any ethical approval (as also mentioned above).
2.3 Interviews
Is it possible to attach the interview guide as an appendix?
The sentence (line 184-185, In the present study we analyse what competences … ) could be moved to the next paragraph about the analysis process.
- Results
I found the result interesting and also easy to read and follow. However, I have some comments and suggestion that could improve and make the result clearer.
In the first paragraph my suggestion is to describe the overall result of the analysis and also explain that the themes are connected to each other and how. I think that it is not only communication that is emphasized in other themes, there are also other underlying meanings that links the themes together. Since the authors have described a theory that will guide the analysis it is also possible to introduce the interpretation of the results with reference to Lindberg and Rantatalo already in the result section. For example, what themes that are related to qualities (skills), and what themes that are related to symbols (appearance, activity in practice), which now is presented in the discussion.
I also suggest to present the themes in a little different order and make a small change in two of the themes. First, present the themes that are related to “skills”: Communicative and coordinating skills, Engagement and advocacy skills, advisory and guidance skills. Then, the all themes that are related to symbols could be named with the verb being: Being emphatic and therapeutic, being responsive and flexible, and being professional and trustworthy. I think that this order presenting the themes matches the theoretical analysis in a better way.
Another suggestion I have, is to change the theme being responsive and flexible to being persistent and flexible, which also creates an interesting tension in the theme. I found that the authors describe, in the abstract and in the discussion (line 419), highlights that being persistent as a finding that not have been recognized before. Therefore, I think it would be an advantage of being persistent also is seen in one of the themes. If the authors agree with this change, they have to move the sentence about being persistent (line 319-321) from the theme of professionality to this theme.
One more comment about the result is to consistently use the term participants instead of using both interviewees and participants.
- Discussion
The first sentence in the discussion is a little unclear. I understand the end of the first sentence more like ….. describe skills and qualities of professional competence that they experience facilitate/support the RTW-process.
Again, the aim of the study has to be consistent with how the aim is described earlier.
Depending on if the theorical analysis is presented here or in the result, the part of the section where the themes is presented has to be changed (line 343-349).
In line 360, the authors write that the findings are in contrast to earlier studies, this is interesting and have perhaps to be clarified in some way to become easier for the reader to understand. One suggestion is to also give examples of how performing knowledge is presented in the result.
The paragraph (line 400-405) could be moved to the section where implications for future studies are mentioned (line 426-432). I think it would be valuable with future studies that investigate needs from patient groups with different diseases and difficulties.
The methodological discussion is quite short (line 423-425). I would like to expand the discussion with some more examples describing strengths and weaknesses of the study, and how the trustworthiness and credibility is reached.
The last paragraph in the discussion is important and summarizes the main result from the study. I found that the study contributes with interesting findings showing how complex the definition of professional competence is, and in relation to the patients’ expectations. It is also interesting that some qualities seem to build on having specific experiences in how to act and practice knowledge. In addition, the study points out the need to be able to tailor interventions with regard to patients’ different needs. In this study the patients with multimorbidity seem to appreciate the specific support they got from the RECO and that this kind of support is a valuable help to able to return to work. The RECO also seem to offer a competence that that no other professionals in the healthcare system have, which further show the need for the competence to avoid the risk of longtime sick-leave.
Author Response
Reviewer 1
Thank you for the opportunity to read the manuscript. I think the study is interesting and a valuable contribution developing the competence of RECOs in the Swedish healthcare system. The manuscript is well written but I have found some things that could improve the manuscript. One important comment is to declare if the study has an ethical approval. I will also suggest a language editing of the manuscript. You find my comments below.
Authors’ response: Thank you for the careful reading and generous comments – we agree that they have improved the manuscript. The manuscript has undergone professional proofreading. Regarding ethical approval, this information is found under the heading “Institutional Review Board Statement”, by the end of the manuscript. There it is stated that the study was conducted according to the guidelines of the Declaration of Helsinki, and was approved by the Swedish Ethical Review Authority (Dnr 2020-00403).
Title: I would like some more information of the study in the title, for example that the participants/patients in the study are persons with multimorbidity and that the context for the study is Sweden. One suggestion I have is: Patients’ with multimorbidity and their view of professional competence for rehabilitation coordinators in the Swedish healthcare system. Or find an appropriate subtitle that could complement with information about the patient group and context the study is made in.
Authors’ response: The title has been changed to include more details about the participants. We do think, however, that it is important that the title reflect that the manuscript concerns coordination of return-to-work process, and that the title is not in risk of being interpreted as concerning coordinating the healthcare system or rehabilitation in general. The title is therefore not exactly as the reviewer suggested.
The aim is described differently in different parts of the manuscript, in the abstract, the introduction and in the discussion. Please look through and decide the aim and then write the aim with the same wording/meaning when it is mentioned.
Authors’ response: The aim has been revised and is now described using the same wording throughout the manuscript.
Another important comment is that I would like the authors to decide, and be consistent when they refer to and write about rehabilitation coordinators. The rehabilitation coordinator has different names depending on which countries and what contexts they operate in. Therefore, it is important to be clear when different notions are used. In the paper, the authors mix different notions throughout the paper, which makes it difficult for the reader to follow. For example, the authors use coordinators, rehabilitation coordinators, RTW coordinators, RECOs, and so on. When the different notions are used, it has to be clear for the reader when and why different notions are used, and also to be consistent in when to use the different notions. For example, when the rehabilitation coordinator is mentioned by the participants, or if the author refers to a Swedish context, RECO could be used.
Authors’ response: Thank you for the careful reading. We have made changes accordingly, and are now more consistent with using RECO as the main term.
Abstract: The abstract may have to be changed after revisions, in this version, the aim is not clearly described.
Authors’ response: We have revised the abstract and clarified the aim.
- Introduction:
In the beginning of the introduction, line 33, after the first sentence, one suggestion is that the authors could explain that most of the studies about coordinators in the RTW process mostly are called RTW coordinators, and include in what context these coordinators are found (for example in workplaces, Europe, worldwide, etc). Thereafter, it could be appropriate to explain the role of the rehabilitation coordinator in Sweden, and introduce the shortening RECO. Another suggestion is to move the sentences about the Swedish context and RECOs to the end of the introduction, to be included in the last paragraph (line 77) before the theoretical framework.
Authors’ response: We have included a sentence explaining that coordinators in the RTW process often are called RTW coordinators in the literature and added what contexts these coordinators are found, before introducing the role of the RECO in Sweden. Hopefully this clarifies the difference and variations in titles, tasks and contexts. We have also clarified when we refer to RTW coordinators in the literature and when we refer to RECOs in order to reduce the use of different notions.
Section 1.1. The theoretical framework is very interesting. But perhaps it could be better to present the theory after the description of RECOs in the Swedish context. In other words, change 1.1 to become 1.2 and vice versa.
Authors’ response: The theoretical framework is now presented after the description of RECOs in the Swedish context, as suggested by the reviewer. We agree that this is better.
Section 1.3 Aim of the study – as mentioned earlier, the authors have to decide and become clearer and more consistent in describing the aim of the study. For example, I perceive that the aim of the study is something like, investigating/exploring how persons with multimorbidity and psychosocial difficulties describe/experience the professional competence of a rehabilitation coordinator (RECO) when needing support in a return-to-work process.
Authors’ response: The aim has been revised and is now described using the same wording throughout the manuscript.
- Method
2.2 Recruitment.
In this section I would like to have a description of inclusion and exclusion criteria. I also lack information if the study has any ethical approval (as also mentioned above).
Authors’ response: We have clarified the inclusion criteria in the first sentence in the sub-heading ‘2.1. Participants’. Information regarding ethical approval is found under the heading ‘Institutional Review Board Statement’, by the end of the manuscript. This is where we understand that the journal prefers this information to appear, as it is an obligatory heading. There it is stated that the study was conducted according to the guidelines of the Declaration of Helsinki, and was approved by the Swedish Ethical Review Authority (Dnr 2020-00403).
2.3 Interviews
Is it possible to attach the interview guide as an appendix?
Authors’ response: The interview guide is in Swedish, and the interview questions does not address competence as such. But we have developed and revised the section describing the interview guide, providing more information about the interview topics.
The sentence (line 184-185, In the present study we analyse what competences … ) could be moved to the next paragraph about the analysis process.
Authors’ response: We noticed that the point with placing this sentence here was not clear. We have revised this sentence, as well as the sentence before, and hope that our intention is clearer now:
A previous study focusing on the more direct answers to the interview questions, by exploring barriers in the RTW due to the participants’ multimorbidity and psychosocial difficulties, and the support received by RECO in different areas of the RTW process is published elsewhere [18]. Regarding the present study, however, there were no specific interview questions addressing professional competence, instead, we explore how the participants during the interviews describe the professional competence of the RECO they encountered.
- Results
I found the result interesting and also easy to read and follow. However, I have some comments and suggestion that could improve and make the result clearer.
In the first paragraph my suggestion is to describe the overall result of the analysis and also explain that the themes are connected to each other and how. I think that it is not only communication that is emphasized in other themes, there are also other underlying meanings that links the themes together. Since the authors have described a theory that will guide the analysis it is also possible to introduce the interpretation of the results with reference to Lindberg and Rantatalo already in the result section. For example, what themes that are related to qualities (skills), and what themes that are related to symbols (appearance, activity in practice), which now is presented in the discussion.
Authors’ response: We agree that the themes are connected to each other, but we cannot see that there are any other underlying meanings connecting one to the other as clear and dominant as communication. We have therefore not elaborated on this further. Regarding how theory have guided the analysis, we have clarified the concepts qualities, skills, symbols, appearance, and activity in practice within themes. We prefer, however, to have a theoretical discussion of the overall findings after the result section.
I also suggest to present the themes in a little different order and make a small change in two of the themes. First, present the themes that are related to “skills”: Communicative and coordinating skills, Engagement and advocacy skills, advisory and guidance skills. Then, the all themes that are related to symbols could be named with the verb being: Being emphatic and therapeutic, being responsive and flexible, and being professional and trustworthy. I think that this order presenting the themes matches the theoretical analysis in a better way.
Authors’ response: Thank you for this nice suggestion. We have now presented the results in line with the suggestion, although we placed “Engagement and advocacy skills” as the third theme because engagement refer to appearance. With this order, we present the themes related to skills first and then those that are related to symbols. The latter has also been renamed as suggested.
Another suggestion I have, is to change the theme being responsive and flexible to being persistent and flexible, which also creates an interesting tension in the theme. I found that the authors describe, in the abstract and in the discussion (line 419), highlights that being persistent as a finding that not have been recognized before. Therefore, I think it would be an advantage of being persistent also is seen in one of the themes. If the authors agree with this change, they have to move the sentence about being persistent (line 319-321) from the theme of professionality to this theme.
Authors’ response: Thank you for this interesting comment! We agree with your reading. The theme “Being responsive and flexible” has been renamed to “Being persistent and flexible” and the sentence about being persistent from the theme of professionality has been moved to this theme instead as well as a quotation from one of the participants (this do not appear by track changes).
One more comment about the result is to consistently use the term participants instead of using both interviewees and participants.
Authors’ response: The term “participant” is now used consistently instead of “interviewee”.
- Discussion
The first sentence in the discussion is a little unclear. I understand the end of the first sentence more like ….. describe skills and qualities of professional competence that they experience facilitate/support the RTW-process. Again, the aim of the study has to be consistent with how the aim is described earlier.
Authors’ response: We agree that the first sentences were a bit confusing. We have revised the first two sentences, and clarified the aim once again, now described using the same wording throughout the manuscript.
Depending on if the theorical analysis is presented here or in the result, the part of the section where the themes is presented has to be changed (line 343-349).
Authors’ response: As we present the theoretical analysis in the discussion and not in the results, no changes have been made to this paragraph.
In line 360, the authors write that the findings are in contrast to earlier studies, this is interesting and have perhaps to be clarified in some way to become easier for the reader to understand.
Authors’ response: We elaborate on the findings being in contrast to previous studies further on in the discussion, where we begin by first stating how it theoretically contrasts in the way we use the term competence and thereafter regarding the empirical findings.
One suggestion is to also give examples of how performing knowledge is presented in the result.
Authors’ response: It is a good suggestion, we have added a sentence giving such examples.
The paragraph (line 400-405) could be moved to the section where implications for future studies are mentioned (line 426-432). I think it would be valuable with future studies that investigate needs from patient groups with different diseases and difficulties.
Authors’ response: We have developed the text at the end of the manuscript about the value of future studies including the perspective of patient groups with different diseases and difficulties.
The methodological discussion is quite short (line 423-425). I would like to expand the discussion with some more examples describing strengths and weaknesses of the study, and how the trustworthiness and credibility is reached.
Authors’ response: We have developed the methodological discussion, now covering 18 lines.
The last paragraph in the discussion is important and summarizes the main result from the study. I found that the study contributes with interesting findings showing how complex the definition of professional competence is, and in relation to the patients’ expectations. It is also interesting that some qualities seem to build on having specific experiences in how to act and practice knowledge. In addition, the study points out the need to be able to tailor interventions with regard to patients’ different needs. In this study the patients with multimorbidity seem to appreciate the specific support they got from the RECO and that this kind of support is a valuable help to able to return to work. The RECO also seem to offer a competence that that no other professionals in the healthcare system have, which further show the need for the competence to avoid the risk of longtime sick-leave.
Authors’ response: Thank you for your reflections!

Reviewer 2 Report
The manuscript covers a very important topic of interest. The manuscript identifies some very interesting themes which require exploring however it does requires some work. Please consider the comments provided as a means of enhancing your manuscript.
Best wishes

Author Response
Reviewer 2
Thank you for your kind invitation to review this paper ‘Patients’ view on important professional competence for coordinators in the return-to-work process’. This paper presents some unique findings in terms of the Swedish context and would contribute to the knowledge in this area; however, it requires substantial rewriting for clarity.
Title: The title could be revisited as the term ‘patients’ does not inform the specific study participants - injured workers. It needs to be more specific and concise.
Authors’ response: The title has been changed to include more details about the participants as suggested by both reviewers. We have clarified in the first paragraph in the introduction that RECOs are not restricted to working with people with work related injuries, as may be the case in other countries. Rather, RECOs work with all types of medical diagnosis resulting in work disability and SA, regardless of what causes the diagnosis.
Abstract: The abstract presents the content of the article, however, has some grammar issues.
Authors’ response: The manuscript has been professionally proofread in order to eliminate any grammatical or spelling errors.
Introduction: The purpose is not clearly explained for the reader. The focus of this study appears to be on the way RECOS interact with the ‘patients’ as opposed to the knowledge they possess. The aim needs to be much more specific and clearer for the reader. Some relevant literature not included.
Authors’ response: The aim has been clarified and is now using the same wording throughout the manuscript. We have also added two more relevant references [14,17].
Methods: The methods lack detail and clarity, making it difficult to replicate. Please refer to comments below.
Authors’ response: We have now added details regarding the choice of method, inclusion criteria, recruitment, the analysis process, the development and content of the interview guide and our approach to the aim with regard to the interview guide. We have clarified that all participants agreed to having the interview recorded and explained the concept meaning bearing units. We have also developed the methodological discussion regarding possible biases.
Results: The findings are provided in detail however some of the comments by the participants do not align with the themes or appear to have been lost in interpretation. For example: Communicative and coordinating skills – this participant alludes to feeling a sense of security, yet the theme is on communication and coordinating skills. Advisory and guidance skills – in this context it appears that participants are being ‘told’ rather than being advised or guided.
Authors’ response: We have added a quotation regarding communication to make the content more align with theme. The quote where the participant describes that the RECOs coordinating skills involves a sense of security is an example why coordinating skills is deemed as valuable. We believe this adds valuable depth to the theme. Regarding the theme advisory and guidance skills, the participants did not use the word “being told” and they did not express feeling forced or told what to do, rather the RECOs gave them their opinion or “supported” or “showed” them how to think or behave, suggestions which they could choose to follow or not.
Discussion: The discussion lacks cohesion due to significant grammatical errors, which prevent the flow of the sentences and paragraphs making it difficult to read.
Authors’ response: The manuscript has been professionally proofread in order to eliminate any grammatical or spelling errors.
Limitations: Refer to comments below
Conclusion: May need revisiting when the aim is refined. The are no future research directions mentioned.
Authors’ response: At the end of the discussion section we have given suggestions for future research, where we suggest that further studies could benefit from more closely investigating if and how the different fields of place of work, thus contextual factors, may affect the competence perceived as important, both from the perspective of patients and the RECOs themselves and that it would be valuable to add the need of different patient groups with different diseases and difficulties.
General feedback: The paper presents some new findings, however there is literature that could have been sighted to support the review (see below). The writing is not consistent and clear, therefore does not logically flow throughout the manuscript.
Authors’ response: The manuscript has been professionally proofread in order to eliminate any grammatical or spelling errors.
The following article may be of interest to the authors: Bohatko-Naismith, J., James, C., Guest, M., Rivett, D.A. and Ashby, S. (2019), "An exploratory study of the injured worker’s experience and relationship with the workplace return to work coordinator in NSW, Australia", International Journal of Workplace Health Management, Vol. 12 No. 2, pp. 57- 70. https://doi.org/10.1108/IJWHM-07-2018-0089
Authors’ response: We appreciate this very relevant article suggestion; we had missed it. We have now included the reference as well as another one in the introduction, as well as in the discussion.
Specific comments:
Line 101 – 102 Patients or participants? Consistency required
Authors’ response: We have changed the word “patient” to “participant” in the sentence to be more consistent.
Line 143 Why did the authors use semi-structured interviews and not focus groups? And why telephone interviews?
Authors’ response: The main purpose with the interviews was to gain individual experiences of participants situation living with multimorbidity and psychosocial difficulties, their individual need of support and descriptions of what support they received from the RECO. Therefore, individual interviews were preferred. Our previous findings are published elsewhere, and we find it difficult to explain this in detail in the present study without making the text more difficult to follow. We have therefore not elaborated around this in the text, but we have added a sentence explaining that telephone interviews were chosen due to the Covid-19 pandemic.
Line 144 – More detail required on why thematic analysis was chosen for this study…
Authors’ response: We have added an explanation for the choice of thematic analysis.
Line 150 – Did the participants have physical injuries and then developed psychological injuries as a result?
Line 153 Not sure what the statement ‘but somatic diagnosis such as cancer also occurred’ means in this context? Was the participant’s cancer work related? Or was it a secondary diagnosis?
Authors’ response: RECOs in Sweden do not only work with work related injuries, as may be the case in other countries. RECOs work with all types of medical diagnosis that result in work disability and SA, regardless of what causes the diagnosis. We have clarified this in the first paragraph in the introduction.
Line 162 – How did the RECOs decide who to send the invitation letters too? Bias?
Authors’ response: We have added the following italics in the method section:
Thirteen RECOs working at primary healthcare, psychiatry or addiction centres, were asked by the project leader (VS) to send a letter about the study to three to six eligible patients each (in total 70 patients), preferably with different genders, backgrounds, and experiences. Due to the Swedish Secrecy Act, it was not possible for the project leader to obtain any information about those invited.
We also elaborate on potential biases in the final discussion.
Line 179 – How where the questions developed for the semi-structured interviews?
Authors’ response: The interview guide was developed based on previous literature covering RTW coordination and rehabilitation coordinators. We have clarified this.
Line 182 – In the RTW what? Process?
Authors’ response: Thank you for the careful reading. The word process was missing and we have added this to the sentence. Also, the manuscript has been professionally proofread in order to eliminate any grammatical or spelling errors.
Line 182 – Not sure why this statement is deemed necessary here– reference (18)?
Authors’ response: As the questions addressed in the interview guide do not reflect the aim and the topic of the present manuscript, we find it necessary to comment that we have already published the results following from the topics addressed from the interview guide. We believe it would raise possible questions not mentioning that the aim of the present study is not directly reflected in the interview guide, and also possible questions about why we have not analyzed the material from the perspectives reflected in the interview guide (thereby the reference). However, we have revised the text, hoping that the new formulations are clearer:
A previous study focusing on the more direct answers to the interview questions, by exploring barriers in returning to work due to the participants’ multimorbidity and psychosocial difficulties, and the support received from RECOs in different areas of the RTW process, is published elsewhere [18]. Regarding the present study, however, there were no specific interview questions addressing professional competence; instead, we explored how the participants during the interviews described the professional competence of the RECO they encountered.
Line 185 – Should this read competencies not competences?
Authors’ response: We have revised this sentence.
Line 186 – Did you advise the participants that the interviews were being recorded?
Authors’ response: We asked the participants for permission to record the interviews, and all accepted.
Line 190 – Why did you use thematic analysis? You need to justify
Authors’ response: We have added an explanation for the choice of thematic analysis.
Line 191 – Please clarify what you mean by ‘meaning bearing units’
Authors’ response: Meaning bearing units refer to sentences and phrases that contain relevant information and content. We have clarified this.
Line 200 – Data saturation does not occur in the data analysis process it does however occur in the interviewing process.
Authors’ response: We have changed this sentence as we agree that it was an unfortunate description. We, in line with Braun and Clarke (2021) agree that data saturation, meaning information redundancy (whether referring to data collection or during an analysis) bears little meaning. Instead, the sentence now includes an explanation of when we found that the analysis of the themes was deemed conceptually deep enough.
Braun, V., & Clarke, V. (2021). To saturate or not to saturate? Questioning data saturation as a useful concept for thematic analysis and sample-size rationales. Qualitative research in sport, exercise and health, 13(2), 201-216.
Line 217 – A quote to support the communication theme would be helpful
Authors’ response: A quote relating to communication has been added.
Line 289 – Check spelling - Okey - Okay
Authors’ response: Thank you for the careful reading. Spelling on the word okay (we use British English) has been corrected and the manuscript has been professionally proofread in order to eliminate any grammatical or spelling errors.
Line 298 – 303 - I am concerned about this paragraph. Such advice regarding the ‘pace’ of the process should be determined by the physician not RECOs. This advice may cause unnecessary delays in the RTW process
Authors’ response: We agree that this could be problematic and we are aware of research pointing at this. But we also know that Swedish physicians often appreciate RECO giving such advice to them, as a large majority of physicians declare that they have no/very little knowledge about the work places, work tasks or work demands – topics that is in focus for RECOs mission. As researchers, we choose not to question the patient’s view about this or discuss this complex issue here, instead we deem it as valuable to present the patient’s view.
Line 306 – present not presenting
Authors’ response: The manuscript has been professionally proofread in order to eliminate any grammatical or spelling errors.
Line 340 – reference required – previous literature?
Authors response: We have given example to the previous literature that our study adds to.
Line 364 – ’we defined competence, as not only the ability of knowing but also the ability of performing knowledge when encounter a patient in practice’ – I am not sure what you are trying to say here? Do you mean the ability to possess the theoretical knowledge and being able to apply the knowledge?
Authors response: Thank you, we see that this sentence needed clarification, and we have revised it.
Line 384 – not qualifications but characteristics/traits
Authors response: We have changed qualifications to characteristics/traits accordingly.
Line 423 – I would suggest to the authors that there are further limitations such as the recruitment process, volunteer bias etc…
Authors response: Thank you for this comment, we have developed the methodological discussion.

Round 2
Reviewer 1 Report
To authors, I have read the revised manuscript and your point-to-point responses and I am very content with the work you have done to improve the manuscript. You have taken care of my comments carefully and I like how you have chosen to solve my suggestions for improvments. For example, I like how you introduce the result and how you have expanded the methodological discussion. I have no further comments.
Reviewer 2 Report
The authors have made a significant improvement to the orginal submission of the manuscript.
One further comment: please move the paragraph numbered 479 -484 from the results section and place it in the methods section.